# Plant Food Allergy Improvement after Grass Pollen Sublingual Immunotherapy: A Case Series

**DOI:** 10.3390/pathogens10111412

**Published:** 2021-10-30

**Authors:** Fabiana Furci, Luisa Ricciardi

**Affiliations:** Department of Clinical and Experimental Medicine, School and Operative Unit of Allergy and Clinical Immunology, University of Messina, 98214 Messina, Italy; fabianafurci@gmail.com

**Keywords:** rhinitis, grass pollen allergy, plant food allergy, sublingual immunotherapy, tolerance

## Abstract

Background: Cross-reactivity between pollens and plant food has been widely described. Pollen extract subcutaneous immunotherapy in patients with pollens and plant food allergy has been shown to improve tolerance not only to inhalant allergens but also to reduce symptoms in patients with various food allergies. Methods: We retrospectively report our experience with 15 female patients with a positive history for moderate, persistent allergic rhinitis due to grass pollen and oropharyngeal symptoms after ingestion of different plant food. These patients followed a five-grass pollen sublingual tablet immunotherapy for three years in a discontinuous pre-co-seasonal scheme. Results: All 15 patients treated with the 5-grass pollen sublingual tablet immunotherapy, taken once daily for 3 years on a 7-month course, showed improved ocular/nasal symptoms, with a reduction in the use of symptomatic drugs (e.g., nasal corticosteroids and H1 antihistamines). After the first seven-month course of immunotherapy, all patients declared a good tolerance to the intake of fruits and vegetables, and in particular, good tolerance to the offending foods. Conclusions: In conclusion, we have observed improvement of both respiratory and plant food allergies after sublingual immunotherapy (SLIT) with a five-grass pollen tablet.

## 1. Introduction

Cross-reactivity between pollens and plant food has been widely described [1,2,3]. It has been reported to cause the onset of allergic reactions in subjects who develop sensitization to both pollens, developing respiratory allergies and allergies to plant food and causing food allergy clinical manifestations which can vary from oral allergic syndrome (OAS) to anaphylaxis [4].Various advanced therapeutic approaches are under evaluation and include biological agents such as anti-IgE, anti-IL-5 and anti IL-4/IL-13 treatments, as these molecules are important in food-induced allergic inflammatory pathogenesis [5,6]. Allergen immunotherapy (AIT) is a disease-modifying treatment for IgE-mediated allergic diseases [7]. Pollen extract subcutaneous immunotherapy (SCIT) in patients with pollen and plant food allergies has been shown to improve tolerance not only to inhalant allergens but also to reduce symptoms in patients with various food allergies [8]. SLIT has been used for pollen-food allergy syndrome, although further investigation is needed in order to address how to treat these multi-allergic individuals [9,10].

The main responsible proteins for pollen and plant food cross-reactivity are usually lipid transfer proteins (LTP) and profilins; however, the allergic response is typically different [11]. LTP manifestations can range from local oropharyngeal symptoms to anaphylaxis, with the most severe reactions related to the presence of a co-factor such as heat, exercise, strong emotions, alcohol intake or nonsteroidal anti-inflammatory drugs (NSAIDs) [12,13]. On the other hand, profilins usually elicit mild symptoms, like oral allergic syndrome (OAS), and no relation has been identified so far with co-factors. This is in part related to the properties of both type of proteins: LTP are acid- and temperature-resistant, so they reach the gut to be absorbed and activate a systemic response, while profilins are labile and so reactions are usually limited and local [14]. In the case of pollens, LTP have been identified in a few of them, while profilins are widely distributed [15]. What still remains unclear for both groups of proteins is the potential cross-benefit of treating pollen allergy with sublingual immunotherapy to benefit the food allergy or vice versa [16,17].

Here we report our experience with 15 female patients who were consecutively evaluated at the Allergy Unit of G.Martino University Hospital in Messina, Italy, with a positive history for moderate, persistent allergic rhinitis due to grass pollen and oropharyngeal symptoms after ingestion of different plant foods. These patients after undergoing allergic screening were prescribed a five-grass pollen sublingual tablet immunotherapy which they followed for three years in a discontinuous pre-co-seasonal scheme.

## 2. Case Series

We report data retrospectively collected from 15 female patients, aged between 19–35 years, with a positive history for moderate/ persistent allergic rhinitis due to grass pollen and oropharyngeal symptoms (e.g., lip, oral and/or palate mucosa itching) after ingestion of different plant foods. Regarding food allergy manifestations, four had symptoms after plant food ingestion alone, six only elicited symptoms after food and exercise, while in five cases, food allergic reactions appeared in association with nonsteroidal anti-inflammatory drugs (NSAID).

All subjects provided written informed consent for processing personal data according to GDPR 2016/679, and patient anonymity was preserved according to the G. Martino Hospital in Messina, Italy, indications. In particular, the study was conducted in accordance with the ethical standards established in the Declaration of Helsinki of 1946.

Sensitization to grass pollen and plant food was investigated by means of skin prick tests (SPT). A standard SPT panel for inhalant allergens was performed, containing the following components: positive control (histamine), negative control (saline), *Alternaria tenuis, Penicillium, Cladosporium, Aspergillus fumigatus, Dermatophagoides pteronyssinus, Dermatophagoides farinae*, cat and dog dander, *Parietaria judaica, Artemisia vulgaris*, olive tree, *Cupressus*, Betula and a grass mix (containing *Dactylis glomerata, Phleum pratense, Anthoxanthum odoratum, Lolium perenne and Poa pratensis)*. Fruit and vegetable sensitization was evaluated with a panel containing strawberry, peach, banana, kiwifruit, apricot, plum, apple, apricot, tomato, zea mays, wheat, almond, peanut, hazelnut, and latex extracts. Prick-by-prick tests were also performed with the offending foods (tomato, peach, kiwi, parsley) in 8/15 patients, as not all of them gave their consent for these tests.

The sensitization profile of all patients is included in Table 1.

The results obtained in SPT were confirmed by ImmunoCAP (Thermofisher) as far as specific IgE to pollens and plant food allergens were concerned. Specific molecular tests for profilins or LTP determination were not evaluated because they were not available. 

After insufficient response to antihistamines and nasal corticosteroids in all 15 patients, treatment was initiated with a five-grass pollen (*Dactylis glomerata, Phleum pratense, Anthoxanthum odoratum, Lolium perenne and Poa pratensis*) sublingual tablet immunotherapy (Stallergenes Greer, Milan, Italy) for the treatment of their allergic rhinitis. 

All patients had never followed pollen allergy immunotherapy before the study.

Treatment was initiated in a hospital setting under medical supervision with a 100 IR tablet; the following day, another 100 IR tablet was administered, followed by 300 IR tablets taken daily, at home, for a 7-month course and repeated for a further 2 years. No adverse reactions were reported. 

Regarding plant food allergies, patients were advised not to associate the intake of plant foods with known co-factors such as heat, exercise, strong emotions, alcohol intake, and NSAIDs to avoid these enhanced reactions. 

## 3. Results

All 15 patients treated with the 5-grass pollen sublingual tablet immunotherapy, taken once daily for 3 years on a 7-month course, showed improved ocular/nasal symptoms, with a reduction in the use of symptomatic drugs (e.g., nasal corticosteroids and H1 antihistamines) after the first 7-month course of 5-grass-pollen immunotherapy.

Furthermore, all patients reported tolerance to the intake of fruits and vegetables including the offending foods. In particular, the occasional ingestion of plant foods, such as almonds, apples, hazelnuts, kiwis, parsleys, peaches, peanuts, tomatoes, zea mays and wheat contained in foods as bread and pasta, which previously had caused oropharyngeal symptoms, was tolerated. The results in terms of tolerance in each single patient are reported in Table 1.

All patients reported no adverse reactions to the five-grass pollen sublingual tablet immunotherapy during all three years of treatment.

Recurrent ingestion of plant food continued to be tolerated throughout the following two years of treatment with the five-grass pollen sublingual tablet immunotherapy.

## 4. Conclusions

Among the most common causes of food allergies in adolescents and adults, in particular in females, are fruit and vegetables, many of which contain proteins such as profilins and LTP which can be, but not necessarily are, associated with pollen sensitization. As there are several cross-reacting proteins between pollens and plant-derived foods, sensitization may occur to various fruits and vegetables in the same pollen-allergic patient [18]_._

AIT is the cornerstone of IgE-mediated respiratory allergy treatment, and is administered as SCIT or SLIT; AIT with food extracts could avoid food allergy reactions and prevent the risk of potentially life-threatening allergic reactions [19]. 

In patients with pollen and plant food allergies, SCIT with pollen extracts has been reported to both improve tolerance to inhalant allergens and to reduce symptoms in patients with various food allergies [8]; in preliminary trials, SLIT with food allergen extracts placed under the tongue to induce desensitization has been shown to improve tolerance, in particular in patients with peanut and peach allergies [20,21].

In our retrospective study a 5-grass sublingual tablet immunotherapy treatment was started in 15 patients with grass pollen rhinitis and oropharyngeal symptoms after eating plant food. The results obtained were of tolerance to the occasional ingestion of the plant food which had caused allergic symptoms before starting AIT with the five-grass pollen extract.

Tolerance was induced, even only after a seven-month course of five-grass pollen tablet sublingual immunotherapy, and persisted during all three years of treatment. 

These data suggest that, in subjects with grass pollen allergic rhinitis and concomitant oropharyngeal plant food allergic reactions, a five-grass pollen sublingual tablet immunotherapy may induce not only allergic rhinitis improvement but also plant food tolerance. This may be a consequence of the cross-reactivity among pollen and plant food allergens [9]. 

Our data have some limitations. We could not characterize the molecular profile of our patients. Food allergic symptoms behaved like profilins, which are also related to grass pollen allergy, and that could justify the cross-reaction with the immunotherapy with a five-grass pollen extract. However, the presence of profilin is not confirmed in that treatment. On the other hand, co-factors have been described with LTP, but have not yet been described with profilins. Additionally, prick-by-prick tests with plant foods were positive, which again indicates LTP as the offending protein. Another limitation is that patients were not provoked with the offending food before allergen immunotherapy with or without the exposure to co-factors such as exercise or NSAID intake. However, it has been reported that provocation tests with food and cofactor can be challenging and not always reproducible [22,23].

In conclusion, we have observed both the improvement of allergic rhinitis and plant food allergy reactivity after a five-grass pollen tablet sublingual immunotherapy. 

These observations must be confirmed with a better profiling of patients to identify the responsible proteins for the cross-reactive effect, as well as with in vitro studies to demonstrate the common mechanism of action between the grass pollen and the food desensitization. New findings in this field could help in the management of complex pollen-food allergic patients, leading to a more personalized medicine approach.

## Figures and Tables

**Table 1 pathogens-10-01412-t001:** Clinical symptoms, diagnosis, and food tolerance after sublingual immunotherapy.

Patient	Symptoms	Co-Factor		Plant Food Involved	SPT Inhalants	SPT Food Allergens	PbP with the Offending Food	Subsequent Tolerance
		Exercise	NSAID					
1	Oral itching	No	Yes	Peach	Grass Mix	Peach, tomato, zea mays	+	Yes
2	Lip and oral itching	No	No	Tomato	Grass mix, olive tree	Tomato, peach, wheat	+	Yes
3	Oral itching + perioral erythema	No	Yes	Parsley	Grass mix, house dust mites	Peach, tomato, wheat, hazelnut, almond, peanut	+	Yes
4	Oral itching	No	Yes	Peach	Grass mix, olive tree	Peach, wheat	+	Yes
5	Lip itching	Yes	No	Hazelnut	Grass mix, olive tree, house dust mites	Hazelnut, peach	Not performed	Yes
6	Lip itching	No	Yes	Hazelnut	Grass mix	Hazelnut, peanut, almond	Not performed	Yes
7	Lip itching	Yes	No	Almond	Grass mix	Almond, wheat, peanut	Not performed	Yes
8	Lip and oral itching	Yes	No	Wheat	Grass Mix	Wheat	Not performed	Yes
9	Lip itching	Yes	No	Wheat	Grass mix, parietaria judaica, house dust mites	Wheat, apple	Not performed	Yes
10	Lip and oral itching	Yes	No	Wheat	Grass mix	Wheat	Not performed	Yes
11	Lip and oral itching	Yes	No	Wheat	Grass mix	Wheat	Not performed	Yes
12	Lip itching	No	No	Tomato	Grass mix	Tomato, Peach	+	Yes
13	Lip and oral itching	No	No	Kiwi	Grass mix, olive tree, parietaria judaica	Kiwi, apple, hazelnut, almond	+	Yes
14	Lip and oral itching	No	Yes	Tomato	Grass mix	Tomato, wheat, peach	+	Yes
15	Lip and oral itching	No	No	Peach	Grass mix, olive tree	Peach	+	Yes

NSAID: Non-steroidal anti-inflammatory drugs; SPT: skin prick test; PbP: prick by prick. +: positive.

## Data Availability

Not applicable.

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
