# Peer review of "Plant Food Allergy Improvement after Grass Pollen Sublingual Immunotherapy: A Case Series"

_pathogens, 2021, doi:10.3390/pathogens10111412_

Round 1

Reviewer 1 Report

I have no additional comments.

Author Response

We kindly thank the Reviewer for the positive comments on the manuscript. We made the text more fluent and easy to comprehend as we have changed some paragraphs.

Reviewer 2 Report

The authors describe in the manuscript the tolerance to oral allergy syndrome (OAS) associated with allergy to grass pollens, after a regimen of sublingual desensitization to grass pollen.

1.- The oral allergic syndrome (OAS) is not well characterized in patients since, as the authors themselves state, molecular studies have not been carried out. Without this study, the demonstration of OAS associated with pollen allergy is inconclusive despite skin test studies.

2.- Neither skin test area nor specific IgE data are presented to corroborate the OAS and allow subsequent assessment of tolerance to plant foods.

3.- It is claimed that after desensitization patients required less medication, but this claim is also not supported by data from a daily medication consumption survey.

4.-It is not stated in the material and methods how it is concluded that patients tolerate plant foods, nor how many times or under what conditions patients ingested the harmful foods.

5.- After the desensitization process, no skin tests sIgE or sIgG4b were performed, or are not presented in the manuscript, that demonstrate la tolerancia a los alimentos vegetales.

With the very serious lack of data demonstrating food tolerance following herbal pollen desensitization is simply speculative. Furthermore, the manuscript does not provide any novel conclusions about the topic presented.

Author Response

  • In the Introduction, in oder to make the text clearer, we have addedd that " SLIT  has been used for pollen-food allergy syndrome but for further investigation is needed in order to address the complex needs of multi-food allergic individuals".We have also addedd a new reference
  • We have introduced in the text that " we report a retrospective study" so that the research design is clearer.
  • As far as the methods are concerned we have outlined that "the study was retrospective" as we had omitted it.
  • In the results we have introduced that "a significant reduction of "ill days" compared to "well days" was also observed in all 5-grass sublingual tablet immunotherapy treated patients during the grass pollen season".
  • In the Conclusions we have introduced that "Our observations suggest that in plant food allergic patients, if grass pollen allergy is also present, prescribing a 5-grass-pollen sublingual tablet immunotherapy, following EAACI (European Academy of Allergy and Clinical Immunology) guidelines on allergen immunotherapy, may induce plant food tolerance, thereby preventing allergic symptoms.  The risk of potentially life-threatening allergic reactions could therefore be avoided".
  • We have finally added a new reference.

Reviewer 3 Report

The contents is not enough, but it is acceptable as a case reports.  I would like to request details why the patients were getting better.

In some patients, antigens of food allergy and grass pollen has not strong cross-reactivity, but SLIT was effective for all patients. It is better to show the reason. 

Perhaps the effects is evaluated by the reduction of subjective symptoms from the patients, however, the evaluation style is not unclear.  It is better to show the details of results.

Author Response

  • We have introduced that the study was retrospective in the introduction and also in the methods as we had omitted it
  • In the results we have addedd that a significant reduction of "ill-days" compared to "well days" was also observed in all 5-grass sublingual tablet immunotherapy treated patients during the grass pollen season.

Round 2

Reviewer 2 Report

The authors have not modified any of the suggestions made in the manuscript. They still do not present any data clearly demonstrating the effect of grass pollen immunotherapy on other plants. The conclusion of the manuscript is speculative.
They simply cite, without providing any data, that the ill days decrease with respect to the good days of the patients, which only refers to the effect on the inhalants, not on the OAS.

Author Response

Dear Editor, Please find enclosed the answers to your report form. We are modified the suggestions made in the manuscript as you requested and are sorry that what we had done was insufficient.

We have reviewed all the manuscript and will answer point to to point.

  • We have improved the introduction by editing as following:

Cross-reactivity between pollens and plant food has been widely described [1-3]. It has been reported to cause the onset of allergic reactions in subjects who develop sensitization to both pollens, developing respiratory allergies, and plant food, causing food allergy clinical manifestations which can vary from Oral Allergic Syndrome (OAS) to anaphylaxis [4].Various advanced therapeutic approaches are under evaluation and include biological agents such as anti-IgE, anti-IL-5 and anti IL-4/IL-13 treatments, as these molecules are important in food induced allergic inflammatory pathogenesis [5-6]. Allergen Immunotherapy (AIT) is a disease modifying treatment of Ig-E mediated allergic diseases [

4.Yagami A, Ebisawa M. New findings, pathophysiology, and antigen analysis in pollen-food allergy syndrome. Curr Opin Allergy Clin Immunol 2019; 19 (3): 218-223.

             5.Yao-Hsu Yang, Bor-Luen Chiang. Novel approaches to food allergy. Clin Rev Allergy Immunol. 2014; 46:250-257.

          6.Rial MJ, Barroso B, Sastre J. Dupilumab for treatment of food allergy. J Allergy Clin Immunol Pract 2019

          7.Halken S, Larenas-Linnemann D, Roberts G., Calderon MA, Angier E, Pfaar O, Ryan D et al. EAACI guidelines on allergen immunotherapy: prevention of allergy. Pediatr Allergy Immunol 2017; 28(8):728-745.

………………………… SLIT has been used for pollen-food allergy syndrome, although further investigations are needed in order to address how to treat these multi-allergic individuals [9,10].

9.Matricardi PM, Kleine-Tebbe J, Hoffmann J, Valenta R, et al. EAACI Molecular Allergology User’s Guide. Pediatr Allergy and Immunol 2016;27 (23) :1-250

10.Incorvaia C, Ridolo E, Mauro M, Russello M, Pastorello E. Allergen immunotherapy for birch-apple syndrome: what do we know? Immunotherapy 2017; 9(15):1271-1278.

  • We have better presented the research design:

Here we report our experience with 15 female patients who were consecutively evaluated at the Allergy Unit of G.Martino University Hospital in Messina, Italy, with a positive history for moderate, persistent allergic rhinitis due to grass pollen and oropharyngeal symptoms after ingestion of different plant foods. These patients after undergoing allergic screening were prescribed a 5-grass-pollen sublingual tablet immunotherapy which they followed for 3 years in a discontinuous pre-co-seasonal scheme.

 Case series

We report data retrospectively collected from 15 female patients, aged between 19-35 years, with a positive history for moderate/ persistent allergic rhinitis due to grass pollen and oropharyngeal symptoms (e.g. lip, oral and/or palate mucosa itching) after ingestion of different plant foods. Regarding food allergy manifestations, 4 had symptoms after plant food ingestion alone, 6 only elicited symptoms after food and exercise, while in 5 cases food allergic reactions appeared in association with nonsteroidal anti-inflammatory drugs (NSAID).

…….. Prick-by-prick tests were also performed with the offending foods (tomato, peach, kiwi, parsley) in 8/15 patients as not all of them gave their consent for these tests.

           …………………. The results obtained in SPT were confirmed by ImmunoCAP (Thermofisher) as far as specific IgE to pollens and plant food allergens were concerned. Specific molecular tests for profilins or LTP determination were not evaluated because not available.

  • In the Results section we have tried to be more clear:

………………….. Furthermore, all patients reported tolerance to the intake of fruits and vegetables including the offending foods. In particular the occasional ingestion of plant food, such as almond, apple, hazelnut, kiwi, parsley, peach, peanut, tomato, zea-mays and wheat contained in foods as bread and pasta, which previously had caused oropharyngeal symptoms, was tolerated. The results in terms of tolerance in each single patient are reported in Table 1.

………………….. Recurrent ingestion of plant food continued to be tolerated throughout the following two years’ treatment with the 5-grass-pollen sublingual tablet immunotherapy.

  • We structured the conclusions in order to make them supported by the results:

……………………… in the same pollen allergic patient [18].

AIT is the cornerstone of IgE-mediated respiratory allergies treatment, administered as SCIT or SLIT; AIT with food extracts could avoid food allergy reactions and prevent the risk of potentially life-threatening allergic reactions [19].

……………………………… In our retrospective study a 5-grass sublingual tablet immunotherapy treatment was started in 15 patients with grass pollen rhinitis and oropharyngeal symptoms after eating plant food. The results obtained were of tolerance to the occasional ingestion of the plant food which had caused allergic symptoms before starting AIT with the 5-grass pollen extract.

Tolerance was induced, even only after a 7 month course of 5-grass pollen tablet sublingual immunotherapy, and persisted during all the 3 year treatment.

These data suggests that in subjects with grass pollen allergic rhinitis and concomitant oropharyngeal plant food allergic reactions a 5-grass-pollen sublingual tablet immunotherapy, may induce not only allergic rhinitis improvement but also plant food tolerance. This may be a consequence of the cross-reactivity among pollen and plant food allergens [9].

We do hope that what we have introduced in the text made the writing and data clearer and more accurate.

Sincerely,

Fabiana Furci,  Luisa Ricciardi

Round 3

Reviewer 2 Report

The authors have resolved some of the issues suggested, but there is no verification of some of their conclusions.